# Reliability and minimal detectable change of the MX3 hydration testing system

Ian Winter[1], Josie Burdin[2], Patrick B. Wilson [3]*

1 Human Performance Laboratory, School of Exercise Science, Old Dominion University, Norfolk, VA, United States of America, 2 School of Rehabilitation Sciences, Old Dominion University, Norfolk, VA, United States of America, 3 Associate Professor of Exercise Science, Human Performance Laboratory, School of Exercise Science, Old Dominion University, Norfolk, VA, United States of America

* pbwilson@odu.edu

## Abstract

Assessing hydration status outside of laboratories can be challenging given that most hydration measures are invasive, stationary, costly, or have questionable validity. This study investigated the within-day, test-retest reliability, and minimal detectable change (MDC) of the MX3 Hydration Testing System (MX3 Diagnostics), a relatively low cost, noninvasive, and portable method to measure saliva osmolality. Seventy-five adults (44 men, 31 women; 29.6±10.8 yr, 171.1±9.2 cm, 79.1±15.4 kg) presented two saliva samples approximately 3 to 5 minutes apart. Fluid intake was avoided for at least 5 minutes prior to sample collections. For each sample collection, a researcher used the MX3 to tap the tip of a test strip to saliva on a participant's tongue. Intra-class correlation coefficients (ICCs) and MDC were calculated as measures of reliability. Means for the two measurements were 64.3±19.0 mOsm and 65.5±17.9 mOsm (mean difference of -1.2±13.0 mOsm; t = -0.806, p = 0.423). Further analysis with a two-one-sided test procedure revealed equivalence between the two SOSM measurements (p < 0.001 for upper and lower bounds). Within-day repeat measures yielded an ICC of 0.75 and an MDC at the 90% confidence level of 21.3 mOsm. With moderate-to-good reliability, the MX3 appears to be a practical choice for reliably measuring moderate-sized changes (>20 mOsm) in saliva osmolality outside of laboratory constraints.

## Introduction

Hypohydration, an inadequate hydration status, can result in both cognitive and physical impairments [1, 2]. Not only can hypohydration impair cognitive and physical function, but reduced body fluid storage can decrease sweat rate [3]. This can be particularly detrimental as evaporative cooling from sweat is the human body's primary means of thermoregulation in the heat [3]. According to a joint position statement from the American College of Sports Medicine, Academy of Nutrition and Dietetics, and Dietitians of Canada, decreases in exercise performance become apparent when hypohydration exceeds 2% of body weight, substantial when fluid losses exceed 5% of body weight, and when fluid losses approach 6–10% of body weight, heat stroke and heat exhaustion can ensue [1].

**Data Availability Statement:** All relevant data are within the manuscript and its Supporting Information files.

**Funding:** The author(s) received no specific funding for this work.

**Competing interests:** The authors have declared that no competing interests exist.

Several methods exist to assess hydration status at the individual level, including plasma osmolality, urine osmolality, urine specific gravity, urine color, changes in body weight, and more [4]. However, no single recognized gold standard of hydration status exists, although plasma osmolality, electrolyte lab values, and blood urea nitrogen to creatinine ratio (BUN:Cr) are frequently utilized as reference methods in clinical settings [5]. The majority of frequently utilized hydration status measures are invasive, stationary, and/or costly, indicating the need for less-invasive, less-costly, portable, reliable, and valid tools. Assessing hydration status outside of laboratories (e.g., field settings) has presented a long-standing challenge [6], but technological advances are allowing some of the barriers around portability and cost to be overcome.

Saliva osmolality (SOSM), which is the number of solute particles per kilogram of saliva, is one hydration biomarker that potentially addresses the concerns around invasiveness, cost, and portability associated with other methods. Muñoz and colleagues [7] assessed several hydration biomarkers during active and passive dehydration and found SOSM to be a sensitive, specific, and valid measure of hydration status during active periods of hypohydration (i.e., physical activity in the heat). These results suggest SOSM can be used alongside commonly used hydration assessment methods, such as changes in body mass and urine specific gravity. Additionally, other studies, conducted in both sport and clinical settings, indicate SOSM may represent a valid and reliable marker for hydration status [5, 6, 8]. While research suggests that SOSM is a viable metric for evaluating hydration status, the commercial development of such devices that measure SOSM has been limited. Recently, however, a handheld portable device that tests SOSM has been developed by MX3 Diagnostics (Austin, TX) and is currently available for purchase.

The MX3 Hydration Testing System (HTS) is a portable, handheld osmometer developed for point-of-care, spot SOSM measurement. Limited research has been published on the MX3 HTS, with little-to-no data on its reliability. However, a small number of studies have utilized the MX3 HTS to detect hypohydration in clinical populations [9, 10]. Given that SOSM has shown more between-person and day-to-day variability than plasma osmolality and urine specific gravity [11, 12], evaluating the reliability of this handheld saliva osmometer under a variety of conditions is important. No known study to date has assessed the within-day, test-retest reliability, and minimal detectable change (MDC) of the MX3 HTS for the assessment of saliva osmolality. The MDC represents the minimum change that, with a specified degree of confidence, can be considered as greater than measurement error [13]. In practical terms, the MDC is useful because it provides a benchmark for researchers and practitioners to determine whether an observed change in a given individual is likely to be true change. Thus, the purpose of this investigation was to determine the within-day, test-retest reliability, and MDC of the MX3 HTS.

## Materials and methods

### Participants

A total of 75 individuals (44 men, 31 women; 29.6±10.8 yr, 171.1±9.2 cm, 79.1±15.4 kg) volunteered to participate in this study. Participants were recruited via convenience sampling through multiple avenues in the Hampton Roads, Virginia area. Individuals were eligible for study inclusion if they were at least 18 years of age, weighed under 330 pounds, did not have an implanted electrical device (i.e., pacemaker), and were not pregnant. The latter three criteria were related to a different aim of the study that involved examining the associations between body composition (measured via bioelectrical impedance) and SOSM. Recruitment and enrollment for the study lasted from May 24th, 2023 until April 30th, 2024.

## Ethics

The protocols and procedures for this original research were submitted to and approved by Old Dominion University's Institutional Review Board (reference #23–058). Prior to study participation and data collection, potential participants went through an informed consent process in which the procedures, inclusion criteria, potential risks and benefits, etc. were explained, and they were given the option to choose to either participate or not. Individuals agreeing to participate provided their written consent. Precautions were taken to ensure participants' identifiable data remained confidential.

## Procedures

A variety of locations were used for assessments given the portability of the MX3 HTS device involved in data collection; locations included the Human Performance Laboratory at Old Dominion University, as well as several local gyms and a weightlifting event. Procedures were not altered based on the location of data collection. After signing an informed consent document, any individuals that had fluids with them were asked to refrain from drinking throughout the remainder of testing, including for at least 5 minutes prior to obtaining the initial saliva sample, due to the impact of recent fluid intake on SOSM [12]. Participants completed a questionnaire inquiring about age, sex, race/ethnicity, exercise/physical activity habits, and recent fluid intake. For exercise/physical activity habits, participants could select between sedentary, lightly active, moderately active, and highly active. Total fluid intake over the past 4 hours was assessed using a fill-in-the-blank option.

Subsequently, participants were asked to generate a fresh saliva sample (i.e., new saliva generated after swallowing all existing saliva) and present the sample on their tongue with an open mouth. Using the MX3 HTS, a researcher then tapped the tip of a test strip to the saliva while the participant's mouth remained open. After 10–15 seconds, the MX3 HTS provided a SOSM value along with a respective hydration status category. Participants were categorized as hydrated if SOSM was <65 mOsm, mildly dehydrated if SOSM was 65–99 mOsm, moderately dehydrated if SOSM was 100–150 mOsm, or severely dehydrated if SOSM was >150 mOsm [14]. All three investigators were involved in the sampling, though one investigator (IW) collected more than 90% of the saliva samples. Two of the researchers were Registered Dietitians, while the third was an undergraduate exercise science student. Using sampling guidelines provided by MX3 Diagnostics, all three investigators practiced using the MX3 on multiple occasions prior to using the device for data collection.

After the first SOSM assessment, participants' heights were measured using a stadiometer and body composition was evaluated using an OMRON Full Body Composition Monitor Scale (HBF-514C; Omron Healthcare, Inc., Lake Forrest, IL), which uses bioelectrical impedance analysis. Before the scan, participants were asked to wipe down the palms of their hands and soles of their feet with a moist wipe. Researchers inputted each participant's height, age, and sex into the HBF-514C, which was followed by a period of approximately 30 seconds in which participants stood on the platform and held on to an attached handle with arms out forward and extended for the duration of the scan.

Next, SOSM was assessed a second time using the same process as the first assessment. Time between the two SOSM assessments was approximately 3–5 minutes, short enough for the investigators to not expect real changes in hydration status.

## Statistics

SPSS software (version 29, IBM, Armonk, NY) and Microsoft Excel (2013 version, Microsoft, Redmond, WA) were used for statistical analyses. Specifically, SPSS was used to calculate

descriptive statistics and intra-class correlation coefficients (ICC), while Microsoft Excel was used to generate standard error of the measurement (SEM) and MDC values. The distribution of variables was assessed by examining histograms and Q-Q plots. The data for the first SOSM measurement had one clear outlier (>4 standard deviations [SDs] from the mean), while the second SOSM measurement values had a relatively normal distribution. Removal of the participant with the outlier value for the first SOSM measurement had no meaningful impact on the results, and therefore the analyses using all participants are presented.

An unpaired t-test was used to determine whether there were any differences in SOSM between those who were sampled in the morning vs. the afternoon. A paired t-test was run to test the null hypothesis that the mean difference between the two measurements was 0. Cohen's *d* values were also calculated, using the average of variances in the denominator. The equivalence of the two SOSM measurements was evaluated via a two-one-sided test (TOST) procedure [15] within jamovi [16], using the paired-samples t-test option. The TOST procedure tests whether the following hypothesis can be rejected: the difference between measurements is at least as extreme as the smallest effect size (bounds) of interest [15]. For this study, the lower and upper equivalence bounds were set at Cohen's $d_z$ of -0.5 and +0.5, respectively [15]. Measurements were deemed to be equivalent if the two one-sided t-tests both showed statistical significance at an $\alpha < 0.05$.

The ICCs and SEM were calculated to assess the reliability of repeated SOSM measurements. The within-day ICC was based on two SOSM measurements using a two-way mixed, absolute agreement, single measures method [17]. MDC values were calculated using the following formula: SEM $\times \sqrt{2} \times$ z-score for the desired confidence interval (CI) [18, 19]. This study employed 80% ($MDC_{80}$) and 90% ($MDC_{90}$) CIs; thus, the formulas used were 'SEM $\times \sqrt{2} \times 1.282$' and 'SEM $\times \sqrt{2} \times 1.645$', respectively. The SEM was calculated as follows: SD x $\sqrt{(1-ICC)}$, where the SD was a pooled value from all SOSM observations.

A Bland-Altman plot was created to evaluate bias between differences in the first and second SOSM measurements and estimate an agreement interval [20]. Within the Bland-Altman plot, horizontal lines were plotted on the y-axis to indicate the mean difference in measurements (bias) and 95% limits of agreement (precision). A regression line was also fitted to the Bland-Altman plot and used to test for proportional bias. In addition, a scatterplot was created with the first SOSM measurement on the x-axis and the second SOSM measurement on the y-axis. A linear regression line and line of identity (x = y) were included to compare the regression with hypothetical perfect agreement. Proportional bias was evaluated based on whether the regression slope's 95% CI included 1.0.

A two-sided p-value $\leq 0.05$ was used as the threshold for determining significance. Descriptive statistics are reported as mean±SD.

## Results

Among the 75 adults who participated, mean body fat percentage was 28.6±8.1%, while mean fluid intake four hours prior to assessment was 0.791±0.596 L among 72 participants (three participants did not provide a value for fluid intake). In terms of activity level, 29 participants described themselves as highly active (38.7%), 35 as moderately active (46.7%), nine as lightly active (12.0%), and two as sedentary (2.7%). Of all participants, four self-reported as Asian or Pacific Islander (5.3%), five as black or African American (6.7%), one as Hispanic or Latino (1.3%), one as other race (1.3%), 58 as white (77.3%), and six as two or more races/ethnicities (8.0%).

The means of the first and second SOSM measurements were 64.3±19.0 mOsm and 65.5 ±17.9 mOsm, respectively (mean difference of -1.2±13.0 mOsm; t = -0.806, p = 0.423, Cohen's

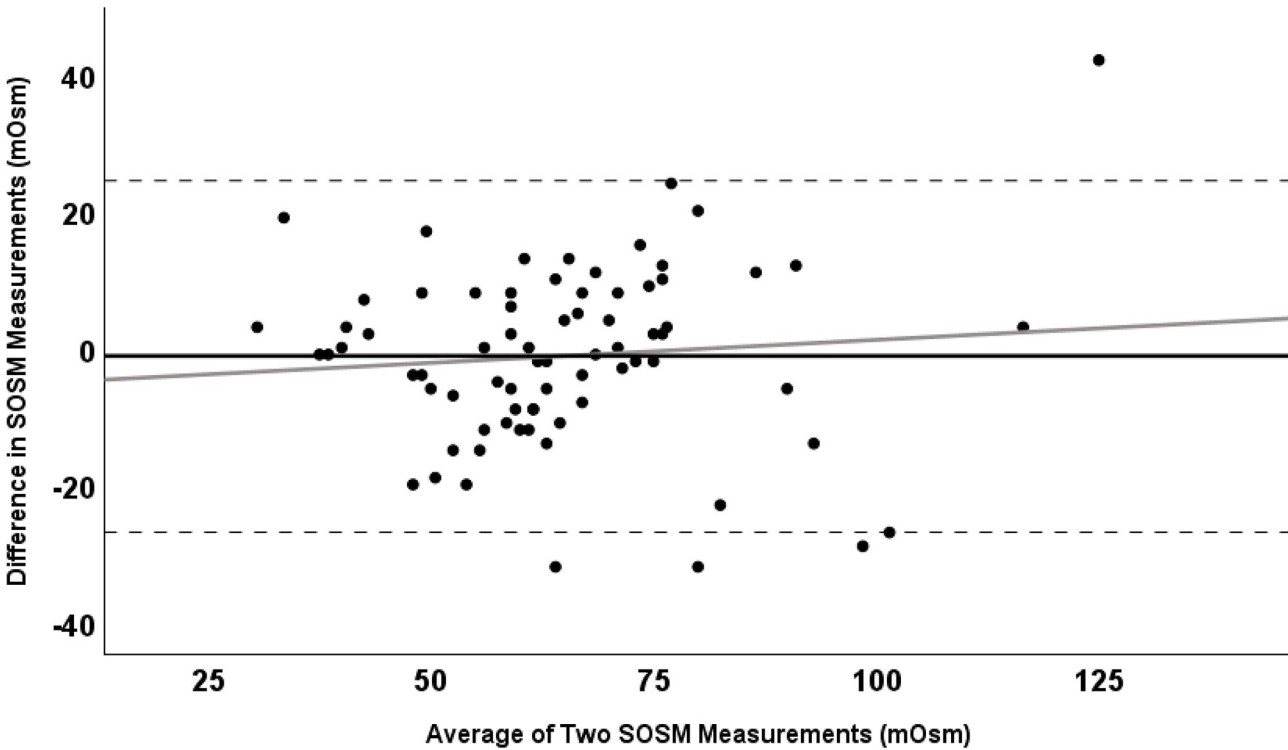

**Fig 1. Bland-Altman plot showing the individual differences in SOSM measurements 1 and 2 by the average of the two measurements.** The mean difference (black), 95% limits of agreement (dashed), and regression (grey) lines are shown.

$d$ = -0.07). The TOST procedure revealed equivalence between the two SOSM measurements (both tests for upper and lower bounds had a p < 0.001).

For the first SOSM measurement, 42 participants were categorized as hydrated (56.0%), 31 as mildly dehydrated (41.3%), and two as moderately dehydrated (2.7%). For the second SOSM measurement, 39 participants were categorized as hydrated (52.0%), 31 as mildly dehydrated (41.3%), and five as moderately dehydrated (6.7%).

Twenty-eight (37.3%) participants had SOSM measurements taken in the morning (before noon), while 47 (62.7%) were sampled in the afternoon. There was no significant difference in average SOSM between those sampled in the morning versus those sampled in the afternoon (65.4±19.7 vs. 64.6±15.8 mOsm; t = 0.203, p = 0.839, Cohen's $d$ = 0.05).

**Fig 1** is a Bland-Altman plot that shows the individual differences between first and second SOSM measurements versus the mean of the two measurements, with horizontal lines depicting 95% limits of agreement and the mean difference between measurements. The regression co-efficient was 0.07 with a 95% CI that covered 0 (-0.11, 0.24), indicating an absence of obvious proportional bias. However, in **Fig 2**, which shows the regression of the first and second SOSM measurements along with a line of identity, there did appear to be some evidence of proportional bias, with a co-efficient of 0.71 (95% CI: 0.56, 0.85) that was different from 1.0.

The ICC, $MDC_{80}$, and $MDC_{90}$ values of SOSM were 0.75, 16.6 mOsm, and 21.3 mOsm respectively. All within-day, test-retest reliability statistics are displayed in **Table 1**.

## Discussion

Although studies have used the MX3 HTS to detect hypohydration [9, 10], to the investigators' knowledge no known study has investigated the reliability of the device for the assessment of

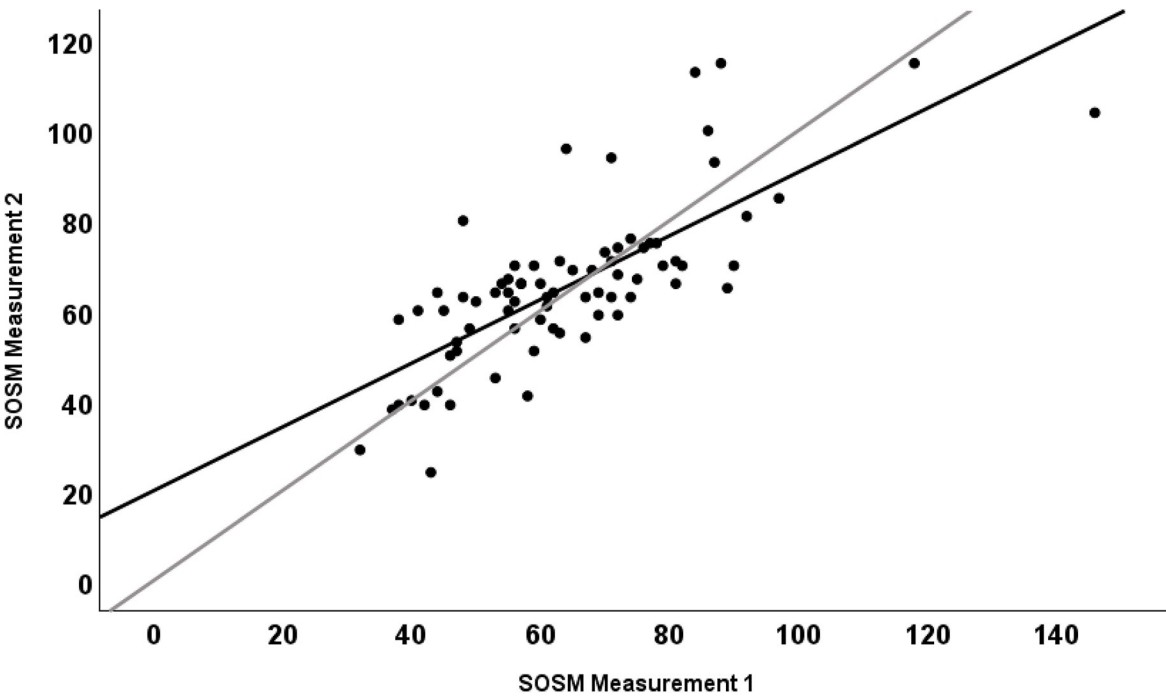

**Fig 2. Scatterplot of the two SOSM measurements with a regression line (black) and a line of identity (grey).**

saliva osmolality. The primary goal of this investigation was to assess the within-day, test-retest reliability of this portable handheld osmometer using multiple spot saliva samples. As ICC is widely used to assess test-retest reliability [17], this was a main outcome measure in determining reliability of the MX3 HTS. Within-day repeat SOSM measures using the MX3 HTS yielded an ICC of 0.75, which corresponds to a moderate-to-good reliability [17]. Based on a $MDC_{90}$ from this ICC, a change in SOSM of 21.3 mOsm or greater would need to be observed in order to be highly confident that this represents true change and not measurement error [13]. Thus, it is with 90% confidence that any repeat measurements that fall within ± 21.3 mOsm of an initial measurement may not represent a real change in SOSM.

It should be noted that a direct comparison of SOSM from two separate spot samples is likely to result in lower test-retest reliability than if both measurements were taken from the same saliva sample (i.e., one saliva sample collected in a sampling tube). While testing of the

**Table 1. Within-day reliability statistics.**

| SOSM measure 1 versus SOSM measure 2 | |
|---|---|
| ICC | 0.75 (95%CI: 0.63, 0.84) |
| SD | 18.4 |
| SEM | 9.2 |
| $MDC_{80}$ | 16.6 |
| $MDC_{90}$ | 21.3 |

CI, confidence interval; ICC, intra-class correlation; SD, standard deviation; SEM, standard error of the measurement; $MDC_{80}$, minimal detectable change at the 80% confidence level; $MDC_{90}$, minimal detectable change at the 90% confidence level.

same saliva sample may provide results more indicative of true device test-retest reliability, the reliability from two separate serial measurements (like those used in the present study) is more realistic in terms of how the MX3 HTS is utilized in field settings (i.e., assess and track changes in hydration status using separate saliva samples over time). Given its minimally invasive procedures, low-cost, and timeliness, taking an average of two back-to-back SOSM readings may be one strategy to improve the reliability of spot samples, as this has been a suggestion in other human assessment reliability studies [21, 22].

There were somewhat contradictory results with regards to evidence of proportional bias. Based on results presented in **Fig 2**, it appears that there may be some proportional bias present, though this was not obvious in the Bland-Altman plot (**Fig 1**). Given that relatively few participants in this study had SOSM values in the moderately or severely dehydrated categories, additional research with larger numbers of participants across the full spectrum of hydration status would be helpful in confirming or refuting the presence of proportional bias. This could involve investigations in which hydration status is experimentally manipulated before serial SOSM measurements are taken.

The investigators of the present study did not assess the validity of the MX3 in comparison to other validated, laboratory-grade osmometers; however, two reports have done so and found the MX3 HTS to be valid against the Osmette III (Precision Systems Inc., Calumet City, IL) and Model 3320 (Advanced Instruments, Norwood, MA) benchtop osmometers [23, 24]. Rather, the values of repeat SOSM measurements were the main interest in the present study, not the respective hydration status categorization provided by the MX3 HTS. With that said, the categorization of hydration status according to SOSM from the MX3 HTS should be interpreted with caution given inter- and intra-individual variation in baseline saliva SOSM [11, 14] and potential interference from recent food, drink, gum, or tobacco consumption [12].

This study is not without limitations. It may have been more ideal for the waiting period between last time of fluid or food consumption and saliva sampling to be longer (e.g., 15 minutes) [12]. Furthermore, if water or a sports drink containing electrolytes and carbohydrates was consumed less than 15 minutes prior to the initial SOSM measurement, then there is a chance SOSM was misclassified [14]. Another limitation is that there was no systematic assessment of exercise in the hours prior to data collection. Some participants were likely to have exercised, which could raise SOSM and theoretically affect the reliability of the MX3. For example, if a person is hypohydrated due to exercise, they may have more difficulty producing saliva consistently [25]. Despite one investigator assessing the vast majority of saliva samples with the MX3 HTS, all three investigators did so at some point during the study, so there is a chance that between-investigator differences in exact sampling technique may have affected SOSM measurements. Future research should address concerns with inter-user reliability and how, if at all, test-retest reliability of the MX3 HTS is altered when repeated measurements are taken immediately after one another instead of with a 3-5-minute gap. Additionally, more research is needed to establish the validity of the MX3 HTS in comparison to other previously validated, laboratory-grade osmometers.

## Conclusions

This study found that the MX3 HTS demonstrates moderate-to-good reliability for the assessment of saliva osmolality. Given its relatively low-cost, noninvasive procedures, portability, near-immediate results, and moderate-to-good reliability, the MX3 HTS appears to be a reasonable and practical choice for reliably measuring moderate-sized changes (i.e., greater than 20 mOsm) in saliva osmolality. These results suggest this tool may be useful for tracking changes in hydration in field settings where hydration status is crucial to performance.

## Supporting information

**S1 Dataset.**
(XLSX)

## Author Contributions

**Conceptualization:** Ian Winter, Patrick B. Wilson.

**Formal analysis:** Ian Winter, Patrick B. Wilson.

**Investigation:** Ian Winter, Josie Burdin.

**Methodology:** Ian Winter, Patrick B. Wilson.

**Project administration:** Patrick B. Wilson.

**Resources:** Patrick B. Wilson.

**Supervision:** Patrick B. Wilson.

**Visualization:** Ian Winter.

**Writing – original draft:** Ian Winter, Patrick B. Wilson.

**Writing – review & editing:** Ian Winter, Josie Burdin, Patrick B. Wilson.

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
