## [Decision Letter · Decision Letter 0]

10 Sep 2024

PONE-D-24-29326Reliability and Minimal Detectable Change of the MX3 Hydration Testing SystemPLOS ONE

Dear Dr. wilson,

Thank you for submitting your manuscript to PLOS ONE. After careful consideration, we feel that it has merit but does not fully meet PLOS ONE’s publication criteria as it currently stands. Therefore, we invite you to submit a revised version of the manuscript that addresses the points raised during the review process.

We look forward to receiving your revised manuscript.

Kind regards,

William M. Adams

Academic Editor

PLOS ONE

Journal Requirements: When submitting your revision, we need you to address these additional requirements. 1. Please ensure that your manuscript meets PLOS ONE's style requirements, including those for file naming. The PLOS ONE style templates can be found at https://journals.plos.org/plosone/s/file?id=wjVg/PLOSOne_formatting_sample_main_body.pdf and https://journals.plos.org/plosone/s/file?id=ba62/PLOSOne_formatting_sample_title_authors_affiliations.pdf

Reviewers' comments:

Reviewer's Responses to Questions

**Comments to the Author**

1. Is the manuscript technically sound, and do the data support the conclusions?

Reviewer #1: Yes

Reviewer #2: Partly

2. Has the statistical analysis been performed appropriately and rigorously? 

Reviewer #1: Yes

Reviewer #2: Yes

3. Have the authors made all data underlying the findings in their manuscript fully available?

Reviewer #1: Yes

Reviewer #2: Yes

4. Is the manuscript presented in an intelligible fashion and written in standard English?

Reviewer #1: Yes

Reviewer #2: Yes

5. Review Comments to the Author

Reviewer #1: Congratulations on your submission. This manuscript assessed the reliability of the MX3 Hydration Testing system among adults. The authors did an excellent job establishing the reliability of this device under the field conditions under which the device is typically used. I have only some minor comments to enhance the quality of the paper:

-Some have argued an equivalence test, such as the two-one-sided (TOST) test is preferable to a t-test in studies that are specifically evaluating equivalence. Rather than running a paired t-test for comparison, I suggest completing the TOST procedure in order to determine statistical equivalence between these repeated measures of the MX3 device.

Lakens D, Scheel AM, Isager PM. Equivalence Testing for Psychological Research: A Tutorial. Advances in Methods and Practices in Psychological Science. 2018;1(2):259-269. doi:10.1177/2515245918770963

-In table 1, can you please also provide the 95% confidence interval for the ICC?

-In the methods can you provide a little more detail on the questionnaire used to determine fluid intake from the four hours prior?

-Did you test for proportional bias in your Bland-Altman analysis?

-Do you have any additional information about the context in which the measurements were taken and if this was related in any way to the agreement? For example, some participants were likely post-workout at the gym or weightlifting event where the data were collected as compared to some other scenarios where a disturbance to hydration status may have been less likely. Can you comment on if you think the context would influence the agreement?

Reviewer #2: Summary and Overall Impression

This study investigates the within-day, test-retest reliability and minimal detectable change (MDC) of the MX3 Hydration Testing System for measuring saliva osmolality. The research addresses an important gap in the literature about the reliability of this portable device for assessing hydration status. Overall, the manuscript is structured well to address the need for confirming the use of the MX3 HTS for field-based hydration assessment and provides valuable information for interested researchers and practitioners. The study's strengths include a clear research question, adequate methods, and highlights practical implications of the findings. However, there are some areas that authors are highly encouraged to consider revising before agreeing on the final draft of the manuscript for publication.

6. PLOS authors have the option to publish the peer review history of their article (what does this mean?). If published, this will include your full peer review and any attached files.

Reviewer #1: No

Reviewer #2: **Yes: **Bahman Adlou

---

## [Author Response · Author response to Decision Letter 0]

30 Sep 2024

Reviewer #1

Comment: Congratulations on your submission. This manuscript assessed the reliability of the MX3 Hydration Testing system among adults. The authors did an excellent job establishing the reliability of this device under the field conditions under which the device is typically used. I have only some minor comments to enhance the quality of the paper:

• Response: We appreciate the positive review and helpful comments from the reviewer. We feel the revisions made, based off their review, have improved the quality of the manuscript. 

Comment: Some have argued an equivalence test, such as the two-one-sided (TOST) test is preferable to a t-test in studies that are specifically evaluating equivalence. Rather than running a paired t-test for comparison, I suggest completing the TOST procedure in order to determine statistical equivalence between these repeated measures of the MX3 device.

Lakens D, Scheel AM, Isager PM. Equivalence Testing for Psychological Research: A Tutorial. Advances in Methods and Practices in Psychological Science. 2018;1(2):259-269. doi:10.1177/2515245918770963

• Response: Thanks for the suggestion. We have carried out a two-one-sided test (TOST), which indicates that there was equivalence between the two SOSM measurements (both tests for upper and lower bounds had a p < 0.001).

Comment: In table 1, can you please also provide the 95% confidence interval for the ICC?

• Response: We have added the 95% CI for the ICC.

Comment: In the methods can you provide a little more detail on the questionnaire used to determine fluid intake from the four hours prior?

• Response: We have provided additional information on the questionnaire, particularly as it relates to the assessment of exercise/physical activity.

Comment: Did you test for proportional bias in your Bland-Altman analysis?

• Response: Thanks for the suggestion. This has been added to the manuscript. We fitted a regression line to the Bland-Altman plot and used it to test for proportional bias. The regression co-efficient was 0.07 with a 95% CI that covered 0 (-0.11, 0.24), indicating an absence of obvious proportional bias. However, in Figure 2, which shows the regression of the first and second SOSM measurements along with a line of identity, there did appear to be some evidence of proportional bias, with a co-efficient of 0.71 (95% CI: 0.56, 0.85) that was different from 1.0. 

Comment: Do you have any additional information about the context in which the measurements were taken and if this was related in any way to the agreement? For example, some participants were likely post-workout at the gym or weightlifting event where the data were collected as compared to some other scenarios where a disturbance to hydration status may have been less likely. Can you comment on if you think the context would influence the agreement?

• Response: We did not systematically assess exercise engagement in the hours prior to data collection. Given that some participants were likely to have exercised in some manner prior to data collection, it may be expected that SOSM would be increased in those individuals (Walsh et al., 2004; https://www.sciencedirect.com/science/article/abs/pii/S0003996903002127). Regardless, two samples were collected only a few minutes apart, so we doubt that reliability is likely to be substantially impacted in this context. That being said, we cannot fully rule out the possibility that the strength of reliability may be impacted by hydration status. For example, if a person is hypohydrated, they could potentially have more difficulty producing saliva consistently. As such, we have added the following text to the limitations section: “Another limitation is that there was no systematic assessment of exercise in the hours prior to the data collection. Some participants were likely to have exercised, which could raise SOSM and theoretically affect the reliability of the MX3. For example, if a person is hypohydrated due to exercise, they may have more difficulty producing saliva consistently [25].”

Reviewer #2

Comment: The rationale for the approximately 5-minute gap between measurements should be better explained. Was this based on previous literature or practical considerations? While the study did not explicitly state the directed use of MX3 HTS for athletic population, it seemed as implied throughout the manuscript. To which athletes does this approx. 5 min gap apply? What age, or competitive level? Was this applied in validation of other field-based measures? 

• Response: This 3-5-minute gap was chosen for practical and methodological purposes–not particularly developed for any type or group of athletes. We wanted this period to be short so that actual biological changes in hydration status would be minimal. Between measurements, body composition was assessed using a portable bioelectrical impedance analysis scale. The resulting 3-5-minute gap between measures was short enough to not expect substantial, real changes in hydration status. When assessing reliability, the period between measurements should be selected so that there is minimal real biological change. 

Comment: The authors sampled a large and heterogeneous population, which is commendable. However, the study does not include an analysis of inter-rater reliability, despite using three investigators. Given the sample size, it would have been feasible to conduct ICC and MDC analyses within each rater or at least assess inter-rater reliability to ensure consistency across different users. This is particularly important since recent literature by Ely et al. (2011) highlighted the limitation of salivary osmolality as a hydration marker, including high inter-individual variability. The manuscript could discuss how the MX3 HTS addresses or accounts for these limitations. 

• Response: The vast majority of measurements (>90%) were carried out by one investigator. While all three investigators used the MX3 device at some point, there is not adequate data from the second and third investigators to assess inter-rater reliability. All three investigators followed a standard sampling protocol recommended by MX3 (https://www.youtube.com/watch?v=bFPtJGFG13c) and practiced using the MX3 on multiple occasions prior to using it for study data collection. The sampling procedure is not particularly complicated, so we do not believe there is likely to be much of an inter-rater effect. Nonetheless, we have suggested this as an area of future research.

Comment: There are concerns regarding the testing protocol. The manuscript does not specify what participants did during the 3-5-minute break between tests, nor whether they participated in strenuous exercise or any physical activity before the tests. Additionally, the time of day when testing was conducted is not described, which could significantly impact saliva osmolality values. Collecting data on diet within four hours prior to testing could have provided insights into how specific food content might affect the accuracy of the measurements.

• Response: We did in fact specify what participants did between the two measurements. As described in the original manuscript, all that participants did during the 3-5-minute gap between measures was step on a BIA device and have their height measured. During that time period, no participant engaged in exercise. 

• Response: While data on diet within four hours leading up to testing could have provided insights into how food choices and content might affect SOSM, this would be unlikely to affect the reliability of the measurements taken and would not, in our minds, justify the additional time/effort required from participants.

Comment: The authors could have considered using mixed models to account for individual differences in their heterogeneous population, such as age, physical activity level, and race. This approach could have provided a more nuanced understanding of how these factors impact the device's reliability and improved the MDC analysis.

• Response: We appreciate that there are many different analytical approaches to use in any given study. However, we feel that it is generally advisable to use the a priori planned analysis to avoid problems that arise from conducting multiple unplanned analyses (e.g., type 1 error increases). Given that a mixed-model approach was not planned prior to the study being conducted, we have reservations about doing that sort of analysis now. In addition, only a small minority of our participants (<10%) were under the age of 50 and <25% were non-white, meaning we wouldn’t have good statistical power to evaluate how those factors impact the results. 

Comment: Some of the references are outdated. While new evidence does not debunk the previous findings, they do make the context more applicable to our current generation of physically active population, such as those in the study.

• Response: A few older references have been switched out for newer ones. We are open to considering any relevant specific suggestions the reviewer may have related to references.

Comment: You could break some of the longer sentences in the first paragraph into smaller segments for better readability. 

• Response: We have broken up a few sentences throughout the manuscript. If the reviewer has additional specific suggestions, we are happy to consider them. 

Comment: Additional context on the importance of field-based hydration assessment in various settings (e.g., sports, military, occupational health) would strengthen the rationale for the study. The population was reviewed in the later segments of the manuscripts, but limited supporting literature were included in either Introduction or Discussion for that. Please consider expanding on the practical implications of the MDC values for different user groups I, or refrain from including them later in the manuscript.

• Response: We have deleted the text that refers to occupational and military settings.

Comment: Clarify whether all participants were assessed at the same time of day, as this could impact SOSM measurements.

• Response: Thanks for the suggestion. We have added an additional analysis (t-test comparing SOSM values between morning and afternoon samples) based on sampling time of day. There was no significant difference in average SOSM between those sampled in the morning versus those sampled in the afternoon (65.4±19.7 vs. 64.6±15.8 mOsm; t=0.203, p=0.839, Cohen’s d=0.05).

Comment: Provide more detail on the training or standardization of the three investigators who used the MX3 HTS. Where are they athletic trainers, dietitians, other allied professionals, or trained research staff? Is the training provided by the device company and was compliance checked?

• Response: Two of the investigators are Registered Dietitians, one being an associate professor of exercise science and the other being a PhD student. The third investigator is an undergraduate exercise science student trained by both dietitians prior to any data collection. Before collecting any study data, all three investigators practiced sampling on multiple occasions according to MX3 HTS guidelines (https://www.youtube.com/watch?v=bFPtJGFG13c). No formal training was provided by the company, as the sampling technique is relatively straightforward (https://www.youtube.com/watch?v=bFPtJGFG13c). 

Comment: Include effect sizes alongside p-values where appropriate to give a better sense of the magnitude of differences.

• Response: Thanks for the suggestion. We have included Cohen d values alongside the t-tests that were carried out. 

Comment: The authors referred to themselves as “the authors of the study” in the beginning, and then referred to themselves as “we.” Review and proofread the manuscript to stay consistent.

• Response: This change has been made.

---

## [Decision Letter · Decision Letter 1]

23 Oct 2024

Reliability and Minimal Detectable Change of the MX3 Hydration Testing System

PONE-D-24-29326R1

Dear Dr. wilson,

We’re pleased to inform you that your manuscript has been judged scientifically suitable for publication and will be formally accepted for publication once it meets all outstanding technical requirements.

Kind regards,

William M. Adams

Academic Editor

PLOS ONE

Additional Editor Comments (optional):

Reviewers' comments:

Reviewer's Responses to Questions

**Comments to the Author**

1. If the authors have adequately addressed your comments raised in a previous round of review and you feel that this manuscript is now acceptable for publication, you may indicate that here to bypass the “Comments to the Author” section, enter your conflict of interest statement in the “Confidential to Editor” section, and submit your "Accept" recommendation.

Reviewer #1: All comments have been addressed

Reviewer #2: All comments have been addressed

2. Is the manuscript technically sound, and do the data support the conclusions?

Reviewer #1: Yes

Reviewer #2: Yes

3. Has the statistical analysis been performed appropriately and rigorously? 

Reviewer #1: Yes

Reviewer #2: N/A

4. Have the authors made all data underlying the findings in their manuscript fully available?

Reviewer #1: Yes

Reviewer #2: Yes

5. Is the manuscript presented in an intelligible fashion and written in standard English?

Reviewer #1: Yes

Reviewer #2: Yes

6. Review Comments to the Author

Reviewer #1: Thank you for addressing all of my comments. I recommend this manuscript be accepted for publication.

Reviewer #2: Congratulations to the authors for conducting this research project and delivering the results to improve the scientific community. There were major revisions done to the manuscripts that could improve the interaction with the data and the findings. Thank you for all your work and attention to detail.

The authors' responses to the revision requests were very appropriate and thorough where applicable. I would recommend (though not necessary) to consider a minor revision:

1. Given that the company has made a great effort to create educational content and tutorial videos for using their device, it could be appropriate to include [in the method] that all users in this study followed the training materials. This serves as both a nod to the industry partner (and others) and a statement on how you opted to standardize the data collection procedure. This method of standardization is becoming more accepted as the industry partners are allocating a bigger budget on training materials.

7. PLOS authors have the option to publish the peer review history of their article (what does this mean?). If published, this will include your full peer review and any attached files.

Reviewer #1: No

Reviewer #2: No

---

## [Editor Report · Acceptance letter]

11 Nov 2024

PONE-D-24-29326R1 

PLOS ONE

Dear Dr. wilson, 

I'm pleased to inform you that your manuscript has been deemed suitable for publication in PLOS ONE. Congratulations! Your manuscript is now being handed over to our production team.

Kind regards, 

on behalf of

Dr. William M. Adams 

Academic Editor

PLOS ONE